# Characterization of Nutrient Intake in Biopsy-Confirmed NAFLD Patients

**DOI:** 10.3390/nu14173453

**Published:** 2022-08-23

**Authors:** Yoshihiro Kamada, Hirokazu Takahashi, Yuji Ogawa, Hideyuki Hyogo, Kyoko Nakamura, Tomomi Yada, Norihiko Asada, Tomomi Bando, Hanako Sawano, Mika Hatanaka, Takako Tosa, Mika Ando, Etsuko Hikita, Kaori Yoshida, Masahiro Koseki, Yoshio Sumida, Kazuhisa Maeda, Makoto Fujii, Shinichi Aishima, Mariko Hayakawa, Atsushi Nakajima

**Affiliations:** 1Department of Advanced Metabolic Hepatology, Osaka University Graduate School of Medicine, Suita 565-0871, Japan; 2Liver Center, Saga University Hospital, Faculty of Medicine, Saga University, Saga 840-8502, Japan; 3Department of Gastroenterology and Hepatology, Yokohama City University School of Medicine Graduate School of Medicine, 3-9, Fukuura, Kanazawa-Ku, Yokohama 236-0004, Japan; 4Department of Gastroenterology, JA Hiroshima Kouseiren General Hospital, Hatsukaichi, Hiroshima 738-8503, Japan; 5Hyogo Life Care Clinic Hiroshima, 6-34-1, Enkobashicho, Minami-Ku, Hiroshima 732-0823, Japan; 6Department of Clinical Nutrition and Dietetics, Faculty of Clinical Nutrition and Dietetics, Konan Women’s University, Kobe 658-0001, Japan; 7Miwa Home Clinic, Tokai 477-0031, Japan; 8Takata Diabetes Internal Medicine Clinic, Kawasaki 210-0817, Japan; 9Akihabara Station Clinic, Tokyo 101-0041, Japan; 10MITO Research Center for Integrative Healthcare, Gifu 502-0013, Japan; 11Medical Corporation Yamashita Hospital, Ichinomiya 491-0913, Japan; 12Sakanoichi Hospital, Sakanoichi, Oita 870-0307, Japan; 13Division of Cardiovascular Medicine, Department of Medicine, Osaka University Graduate School of Medicine, Suita 565-0871, Japan; 14Division of Hepatology and Pancreatology, Department of Internal Medicine, Aichi Medical University, Nagakute 480-1103, Japan; 15Longwood Maeda Clinic, Suita 565-0874, Japan; 16Division of Health Sciences, Osaka University Graduate School of Medicine, Suita 565-0871, Japan; 17Department of Pathology and Microbiology, Faculty of Medicine, Saga University, Saga 840-8502, Japan; 18Department of Health and Nutrition Sciences, Faculty of Human Life Sciences, Nagoya University of Economics, Nagoya 484-0047, Japan

**Keywords:** biopsy-confirmed NAFLD, biases in nutritional intake, vitamin A intake, copper intake

## Abstract

**Objectives:** Weight loss improves the liver pathophysiological status of nonalcoholic fatty liver disease (NAFLD) patients. However, there are few studies that investigate the accurate relationships between nutritional intake and disease progression in NAFLD patients. **Methods:** A total of 37 biopsy-confirmed NAFLD patients were enrolled in this study. Clinical and nutritional control data of 5074 persons were obtained from the National Institute of Health and Nutrition. Each NAFLD subject recorded dietary intake for seven consecutive days using a dietary questionnaire and photographs of each meal. A dietitian analyzed and quantified the nutritional data in each patient. We further analyzed the nutritional intake of NAFLD patients in three groups according to the following criteria: (1) liver fibrosis degree (advanced, early), (2) gender (male, female), and (3) body mass index (BMI) (high, low). **Results:** Excesses or deficiencies of multiple nutrients were found in NAFLD patients compared with control subjects. In addition, there were variations in nutritional intake. (1) The intake of vitamins A, B_6_, and E, pantothenic acid, soluble dietary fiber, and salt was lower in the advanced fibrosis group than in the early fibrosis group. (2) Fat intake was higher in male patients, and dietary fiber intake was lower in both male and female patients compared with control subjects. (3) Saturated fatty acid intake was higher, and copper and vitamin E intakes were lower in patients with high BMI than with low BMI. **Conclusions:** Our study demonstrates that differences were found in some nutrient intake of NAFLD patients and controls and according to the severity of the conditions (liver fibrosis degree, BMI).

## 1. Introduction

Nonalcoholic fatty liver disease (NAFLD) is among the most common causes of chronic liver disease worldwide and is a growing medical problem in industrialized countries [1]. A wide spectrum of histological changes has been observed in NAFLD, ranging from nonalcoholic fatty liver (NAFL), which is generally non-progressive, to nonalcoholic steatohepatitis (NASH). A proportion of patients with NASH develop cirrhosis and hepatocellular carcinoma [2]. Approximately 30% of the general population has NAFLD, and up to 5% of this population has NASH [3,4]. The prevalence of NAFLD is reported to be 20–40% in Western countries and 12–30% in Asian countries [5,6]. Annual health checkup data show that 9–30% of Japanese adults have ultrasonography (US)-diagnosed NAFLD, and NASH is diagnosed in 10–20% or more of NAFLD cases [5,6,7]. There are more than 20 million NAFLD patients in Japan, and it is feared that this number will increase in the future [8,9].

Lifestyle modification, including reduction of calorie intake and/or increased physical activity to achieve weight loss, should be advised for all NAFLD patients [10,11,12]. Weight loss induces liver fat reduction and improves the liver pathophysiological status of NAFLD patients. NAFLD is an independent risk factor for cardiovascular disease (CVD) and cancer of the liver and other organs. Weight loss improves type 2 diabetes mellitus, dyslipidemia, and hypertension. However, whether weight reduction can improve cancer development is unknown. Loss of >=5% total body weight (TBW) improves hepatic steatosis, >=7% TBW induces histological NASH resolution, and >=10% TBW leads to liver fibrosis regression [13,14,15]. In revised Japanese NAFLD guidelines, >=7% weight reduction is also recommended for obese NASH patients [16]. However, the achievement rates for these reductions are as low as 30%, 18%, and 10%, respectively [15]. To achieve weight loss, various kinds of medical and multidisciplinary methodology should be combined to support patient behavior change [17]. The TBW reduction rate in non-obese NAFLD patients [body mass index (BMI) < 25] is considered to be lower than in obese NAFLD patients. Half of patients have been shown to achieve NAFLD remission (defined as an intrahepatic triglyceride content below 5.0% by proton-magnetic resonance spectroscopy) with 3–5% weight reduction within 12-months of lifestyle intervention in non-obese NAFLD patients, and the same was achieved with 7–10% TBW reduction in obese NAFLD patients [18].

Although there is a wealth of information about NAFLD worldwide, there is no study that investigates the accurate relationships between nutritional intake and disease progression in biopsy-confirmed NAFLD patients in Japan. In this study, we conducted an observation study that elucidates the accurate nutritional intake using photographs of meals and nutrition interviews in biopsy-confirmed NAFLD patients.

## 2. Materials and Methods

### 2.1. Ethical Committee Approval

The protocol and informed consent were approved as a multicenter study by each of the following institutional review boards; Osaka University Hospital, Saga University Hospital, JA Hiroshima Kouseiren General Hospital, and Yokohama City University Hospital. Written informed consent was obtained from all subjects at the time of liver biopsy or enrollment in each institute, and the study was conducted in accordance with the Helsinki Declaration.

### 2.2. Study Subjects

A total of 37 patients of biopsy-confirmed NAFLD patients were enrolled in 2020 during the registration period in this study from three hepatology centers in Japan; namely, Saga University Hospital, JA Hiroshima Kouseiren General Hospital, and Yokohama City University Hospital. All biopsy-confirmed NAFLD patients in this study had undergone a percutaneous liver needle biopsy. The indication of liver biopsy was routinely performed for the diagnosis of liver histology in each center. In order to elucidate the recent nutritional intake of NAFLD patients, we set the study registration period to six months. The biopsied liver samples were embedded in paraffin blocks according to standard procedures and stained with hematoxylin and eosin and Masson’s trichrome stains. All biopsy specimens were centrally evaluated by an experienced pathologist (S.A.) who was blinded to the clinical data. Adequate liver samples were defined as > 1.5 cm long and/or having more than six portal tracts. NASH was confirmed according to Matteoni’s classification [19]. NAFLD patients with ballooning hepatocytes (Matteoni type 3) and NAFLD patients with liver fibrosis (Matteoni type 4) were placed in the NASH cohort. Patients whose liver biopsy specimens showed simple steatosis or steatosis with non-specific inflammation were placed in the NAFL cohort. Samples were also investigated and quantified according to NAFLD activity scoring (NAS) [20]. Steatosis (0–3), lobular inflammation (0–2), and hepatocellular ballooning (0–2) were quantified. The individual parameters of fibrosis were scored independently according to the NASH Clinical Research Network scoring system [20]. Early fibrosis was classified as a stage 0–2 (F0–2) and advanced fibrosis was classified as a stage 3–4 (F3–4) disease. The exclusion criteria for this study included a history of other hepatic diseases, a substance abuse-induced hepatic disorder, and a history of alcohol abuse (defined as >20 g of alcohol daily). Based on the median BMI of 30.3, NAFLD patients were divided into two groups; namely, the BMI Low group (BMI <=30.3) and the BMI High group (BMI > 30.3).

Clinical and nutritional data of 5074 with gender and age in common with the case group, obtained from the National Institute of Health and Nutrition in 2019 were used in the analysis (https://www.nibiohn.go.jp/eiken/kenkounippon21/eiyouchousa/koumoku_shintai_chousa.html (accessed on 2 July 2021)). These data were used as the control group.

### 2.3. Study Design

This was a cohort study that prospectively followed the nutritional intake of subjects with a confirmed diagnosis of NAFLD.

### 2.4. Nutritional Analysis Method

Each subject recorded dietary intake of nutrients for seven consecutive days using a dietary questionnaire and photographs of each diet. A dietician analyzed and quantified the nutritional data collected by each NAFLD patient. For the food ingredients list frequency survey sheet used (Figure 1a, Appendix A), we created and used our own sheet for each meal that allows the names of commonly eaten ingredients to be checked based on the classification of the food composition table. 

For the evaluation of nutrient intake, the Food Composition Database of the Ministry of Education, Culture, Sports, Science and Technology (MEXT) was used in this study. Data from photographs taken by patients of everything they ate and drank, with a ruler placed as a guide to determine size (Figure 1b), were checked with the food ingredients list frequency survey sheet, which was checked at each meal, and nutritional calculations were made using the Ministry of Education’s food database. In order to ensure a reliable understanding of what was eaten, the photographs of each food and the food ingredients list frequency survey sheet were used together in this study.

### 2.5. Anthropometry and Laboratory Measurements

Anthropometric variables (height and weight) were measured in the standing position, and BMI was calculated as weight (in kg) divided by the square of height in meters (m^2^). Serum biochemical variables [aspartate aminotransferase (AST), alanine aminotransferase (ALT), γ-glutamyltransferase (GGT), total cholesterol (T-Cho), triglyceride (TG), total protein (TP), albumin (Alb), iron (Fe), creatinine (Cre), hemoglobin A1c (HbA1c), red blood cell count (RBC), and platelet count (Plt)] were measured with a conventional automated analyzer. 

### 2.6. Statistical Analysis

Two analyses were used: an internal comparison within NAFLD patients and an external comparison between NAFLD patients and the Nutrition Examination Survey control population. Continuous variables are summarized as mean and standard deviation (S.D.), and categorical variables are summarized as the number and proportion. The Welch’s t-test and Pearson chi-square test or Fisher exact test were used as appropriate for internal comparison within NAFLD patients. The Welch’s t-test was used to examine differences in nutritional intake status between NAFLD patients and Nutrition Examination Survey individuals. Because individual information was not available for the nutrition survey, t-values and degrees of freedom were calculated based on aggregate information of mean, standard deviation, and number of subjects as shown in the formula below [21,22].
$$t=\frac{\left({\bar{x}}_{NAFLD}-{\bar{x}}_{control}\right)-\left(\mu_{NAFLD}-\mu_{control}\right)}{\sqrt{\left(\frac{s_{NAFLD}^{2}}{n_{NAFLD}}\right)+\left(\frac{s_{control}^{2}}{n_{control}}\right)}}$$
$$v=\frac{\left[\left(\frac{s_{NAFLD}^{2}}{n_{NAFLD}}\right)+\left(\frac{s_{control}^{2}}{n_{control}}\right)\right]}{\left[\frac{1}{\left(n_{NAFLD}-1\right)}{\left(\frac{s_{NAFLD}^{2}}{n_{NAFLD}}\right)}^{2}+\frac{1}{\left(n_{control}-1\right)}{\left(\frac{s_{control}^{2}}{n_{control}}\right)}^{2}\right]}$$
$$H_{0}:\;\mu_{NAFLD}=\mu_{Control}\;\mu :mean,\;n:\mathrm{number}\;\mathrm{of}\;\mathrm{participant},\;\;s^{2}:\mathrm{variance},\;v:\mathrm{degree}\;\mathrm{of}\;\mathrm{freedom}$$

Two-sided *p* values of <0.05 were considered statistically significant. All data were statistically analyzed using JMP statistical software (version 16.1; SAS Institute Inc., Cary, NC, USA).

## 3. Results

### 3.1. Clinical and Nutritional Character of NAFLD Patients Compared with Control Subjects

The anthropometry and laboratory measurement data in NAFLD patients and control subjects (BMI, blood RBC, HbA1c, AST, ALT GGT, and Fe) were higher in NAFLD patients than in control subjects, and Plt and T-Cho were lower in NAFLD patients than in control subjects (Table 1). The histological evaluation data of biopsy-confirmed NAFLD patients revealed that the number of NAS ≥ 4 was 24, and the number of advanced fibrosis ≥ 3 was 14 (Table 2).

The nutritional intake data of K, Zn, Cu, Vitamin (Vit) A, Vit K2, niacin, folate acid, insoluble dietary fiber, and total dietary fiber intake in NAFLD patients was significantly lower than in control subjects. In contrast, Vit E and soluble dietary fiber intake were higher in NAFLD patients than in control subjects (Table 3). We compared our study subject data (control, NAFLD patients) with recommended nutrient intake in Japanese guidelines for nutrient intake (https://www.mhlw.go.jp/stf/newpage_22536.html (accessed on 2 July 2021)). Even in control subjects, the nutrient intake volume was a little smaller than recommendation nutrient intake in some nutrients. In NAFLD patients, nutrition intake was more deficient or excessive than control subjects.

### 3.2. Clinical and Nutritional Characters Divided by Fibrosis Progression

The female/male ratios were 12/11 in the early fibrosis group (stages 0–2) and 8/6 in the advanced fibrosis group (stages 3 & 4). As shown in Table 4, BMI, HbA1c, and T-Cho were lower in the advanced fibrosis group than in the early fibrosis group. 

Total calorie intake was not significantly different between two groups (Table 5). The intake of lipids, protein, Ca, P, Fe, Cu, saturated fatty acid (SFA), and mono-unsaturated fatty acid (mUSFA) was significantly lower in the advanced fibrosis group than in the early fibrosis group. Similarly, the intake of Vit A, Vit B6, Vit E, pantothenic acid, soluble dietary fiber, and salt was also lower in the advanced fibrosis group than in the early fibrosis group.

### 3.3. Clinical and Nutritional Characters Divided by Gender

In males, BMI, blood RBC, HbA1c, AST, ALT, GGT, and Fe values were significantly higher in NAFLD patients than in control subjects (Appendix A). In females, Plt was lower and BMI, blood RBC, HbA1c, T0Cho, AST, ALT, GGT, and Fe values were significantly higher in NAFLD patients than in controls (Appendix A). 

Total calorie intake was not different between male NAFLD patients and control subjects. Fat intake was higher, and Vit A, niacin, and insoluble dietary fiber intakes were significantly lower in male NAFLD patients than in male controls (Appendix A). Total calorie intake was also not different between female NAFLD patients and control subjects. However, Na, Zn, Cu, Vit B2, Vit K2, niacin, insoluble and total dietary fiber, and salt intakes were significantly lower in female NAFLD patients (Appendix A). 

### 3.4. Clinical and Nutritional Characters Divided by BMI

Comparison of clinical data divided by BMI (BMI low group, BMI high group) are shown in Table 6. The nutritional intake data of NAFLD patients in low and high BMI groups are shown in Table 7. Total calorie and Na intakes tended to be higher in the BMI High group than the BMI Low group. Fat and SFA intakes were significantly higher in BMI High group. 

The ratios of each nutrient to total caloric intake, examined to determine the bias in nutrient intake, are shown in Table 8. The SFA intake ratio was higher in the BMI High group than in the BMI Low group. Interestingly, several nutrient intake ratios were lower in the BMI High group than in BMI Low group (K, Mg, Fe, Cu, Vit E). 

## 4. Discussion

The results of this study show an excess or deficiency of multiple nutrients in NAFLD patients compared with control subjects. We compared the nutritional intake data with Japanese dietary guidelines for nutrient intake (https://www.mhlw.go.jp/stf/newpage_22536.html (accessed on 2 July 2021)). There are excesses and deficiencies compared with recommendation in nutrition intake of NAFLD patients in our study. In male NAFLD patients, lipid intake was higher than in control male subjects. In female NAFLD patients, zinc and copper intakes were lower than in control female subjects. In both male and female NAFLD patients, niacin and dietary fiber intakes were lower than in controls. In NAFLD patients with high BMI, total intake calories and SFA intake were higher than in NAFLD patients with low BMI. Interestingly, according to the nutrient intake corrected by total caloric intake, SFA intake was higher, and copper and Vit E intakes were lower in NAFLD patients with high BMI than in patients with low BMI. These findings indicate that NAFLD patients have an unbalanced nutritional intake, a fact that may contribute to the progression of NAFLD. 

Underreporting is a common issue in dietary investigations [23], and might have effect on our study. For example, total calorie intake was not significantly different between NAFLD patients and control subjects in this study. This might be due to underreporting of study subjects. Although this issue would present, we analyzed based on the obtained data in this study, and we found several nutritional intakes were lower in NAFLD patients. Deficiency of Vit A and copper intakes were lower in NAFLD patients than controls, and even lower in patients with advanced fibrosis NAFLD patients. Interestingly, copper and Vit A intakes were further decreased in advanced liver fibrosis NAFLD patients than in early fibrosis NAFLD patients. Liver is the main storage organ of Vit A in animals and human [24,25], and hepatic stellate cell (HSC) is the main storage site of Vit A [26]. Quiescent HSCs store vitamin A in lipid droplets, and activated HSCs lose lipid droplets containing Vit A and produce extracellular matrix proteins, leading to liver fibrosis [27]. Our study demonstrates that Vit A intake was lower in NAFLD patients than in control subjects and further decreased in advanced NAFLD patients. This finding indicates that lower Vit A intake would lead to liver fibrosis progression via hepatic Vit A deficiency in NAFLD patients. Our study also demonstrated Vit E intake was lower in advanced liver fibrosis NAFLD patients than in early fibrosis NAFLD patients. The effects of Vit E on NAFLD were demonstrated in the PIVENS (Pioglitazone vs Vitamin E vs Placebo for Treatment of Non-Diabetic Patients with Nonalcoholic Steatohepatitis) trial [28]. The efficacy of vitamin E on NAFLD/NASH is mainly due to its antioxidant effects. Lower intake of Vit E in advanced fibrosis patients would enhance disease severity in NAFLD patients. 

Copper is a critical part of several cellular processes, including antioxidant function [29]. Cu/Zn superoxide dismutase (SOD1) reduces radical oxygen species by the redox cycling activity of copper in its catalytic center. In our study, not only copper, but also zinc intake was lower in NAFLD patients than in controls. In particular, both copper and zinc intake were significantly reduced in female NAFLD patients compared with in control female subjects. These findings indicate that the deficiency of copper and/or zinc could lead to NAFLD progression. 

Niacin and dietary fiber intakes were also lower in NAFLD patients than in controls in this study. Gender analysis revealed that these intakes were decreased in both male and female NAFLD subjects compared with male and female control subjects. Niacin prevents hepatic steatosis progression in rodent models [30,31]. Niacin also inhibits fatty acid flux from adipose tissue to liver, resulting in the reduction of hepatic triglyceride synthesis and an increase hepatic lipid oxidation. A recent lifestyle intervention study using ^1^H-MR spectroscopy demonstrated that high niacin intake decreases liver fat in a dose-dependent manner [32]. Dietary fiber intake could reduce NAFLD risk [12]. Another recent study demonstrated that increasing consumption of dietary fiber might reduce the risk of NAFLD and its progression [33]. In a Chinese study, dietary intake of total dietary fiber was shown to reduce NAFLD risk in a dose-dependent manner [34]. In our study, dietary fiber intake, especially insoluble dietary fiber intake, was lower in NAFLD patients than in controls. This trend was stronger for females than males, and the total dietary fiber intake was also lower in female NAFLD patients. 

Our analysis according to BMI demonstrates that total calorie and cholesterol intakes tended to be higher in the BMI High group than in the BMI Low group and that the intake of fat, especially SFA, was significantly higher in the BMI High group than the BMI Low group. Comparison of the ratio of each nutrient to total caloric intake, to examine biases in nutritional intake, revealed that several nutrient intake ratios were lower in the BMI High group. Interestingly, intake ratios of copper and Vit E were significantly lower in the BMI High group than the BMI Low group. These findings indicate that uneven nutritional intake could promote the progression of NAFLD. In addition, we compared the nutritional intake between NAFLD subjects with or without type 2 diabetes mellitus (T2DM). T2DM was diagnosed as HbA1c ≥ 6.5%, fasting blood sugar ≥ 126 mg/dL, or treatment with anti-diabetic drugs according to the Japanese clinical practice guideline for diabetes [35]. Among 37 NAFLD patients, 21 subjects were diagnosed as T2DM. We compared the nutritional intake in NAFLD patients with or without T2DM. There are no significant differences between the two groups (data not shown). T2DM had no effects on the nutritional intake in our study subjects. 

Our study has several limitations. First, due to the short time period of the nutritional intake survey (1 week), it is unclear whether it accurately reflects the patients’ usual nutritional intake. Second, the number of NAFLD patients in our study was relatively small. Third, we did not measure patatin-like phospholipase domain-containing protein 3 (*PNPLA3*) gene polymorphism, which is more common in Asian than in Western populations [9]. This gene polymorphism has homozygous mutations in approximately 20% of the general Japanese population [36] and is associated with NAFLD onset and progression [37,38]. Forth, the registration period of our study was during COVID (coronavirus disease)-19 pandemic and the data of control subjects were collected before pandemic. Recently, we investigated the lifestyle changes between before and during COVID19 pandemic [39]. In the study, we found new metabolic dysfunction-associated fatty liver disease (MAFLD) diagnoses and daily alcohol intake increased during pandemic. Although NAFLD patients in the present study did not increase alcohol intake, there might have some additional effects of pandemic on our study results. Fifth, our control data were not collected with the same questionnaire, nor our control subjects have not taken pictures. Since the survey method is to sequentially weigh and record dietary contents rather than having the researcher interview and recall them, we think the survey method is equivalent to taking a photograph of each meal. The National Health and Nutrition Survey (NHNS) used in this study as a control group for comparison includes not only nutritional intake but also lifestyle habits and health examinations (blood data) as used in many other studies [40,41]. This survey, which uses cluster sampling to select and conduct a full survey, has more external validity than using healthy volunteers as the control group because it includes a wider range of subjects, including latent conditions. Despite these limitations, a relationship between inadequate intake of some nutrients and NAFLD disease progression was noted. 

## 5. Conclusions

In conclusion, our study demonstrates that differences were found in some nutrient intake of NAFLD patients and controls and according to the severity of the conditions (liver fibrosis degree, BMI). It is worth noting that copper and Vit A intakes were particularly low in patients with NAFLD. 

## Figures and Tables

**Figure 1 nutrients-14-03453-f001:**
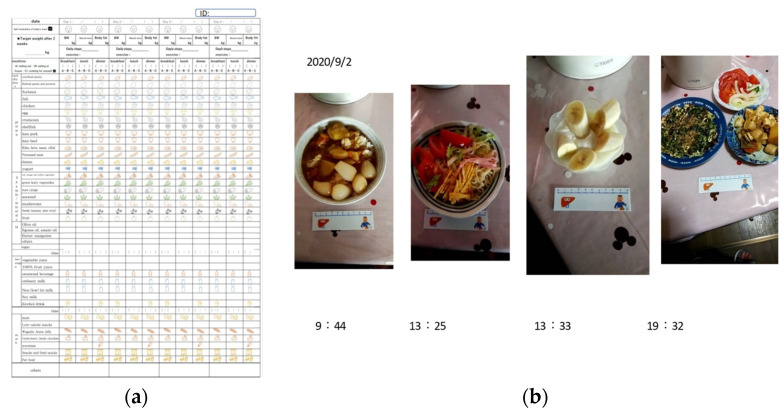
The food ingredients list frequency survey sheet and sample photographs of meals used in this study. (**a**) The food ingredients list frequency survey sheet; (**b**) Sample photographs of NAFLD patients in one day.

**Table 1 nutrients-14-03453-t001:** Comparison of clinical data (NAFLD group, control group).

Parameter	NAFLD (*n* = 37)	Control (*n* = 37)	*p* Value
Age (y.o.)	55.2 ± 10.9	55.2 ± 10.9	n.s.
BMI	29.8 ± 4.7	23.2 ± 3.7	0.0000
SBP (mmHg)	128.6 ± 13.9	131.5 ± 18.1	n.s.
RBC (×10^4^/mm^3^)	457.7 ± 30.3	450.3 ± 48.1	0.0021
Plt (×10^4^/mm^3^)	20.8 ± 8.4	24.1 ± 6.0	0.0247
TP (g/dL)	7.4 ± 0.5	7.3 ± 0.4	n.s.
Alb (g/dL)	4.25 ± 0.38	4.35 ± 0.30	n.s.
HbA1c (%)	6.6 ± 0.9	5.8 ± 0.7	0.0000
T-Cho (mg/dL)	191.0 ± 28.1	206.7 ± 37.1	0.0068
TG (mg/dL)	203.9 ± 166.0	145.7 ± 102.1	n.s.
AST (U/L)	48.8 ± 27.4	24.7 ± 10.3	0.0000
ALT (U/L)	54.3 ± 35.9	22.8 ± 15.1	0.0000
GGT (U/L)	111.5 ± 115.1	37.0 ± 44.5	0.0001
Cre (mg/dL)	0.64 ± 0.18	0.80 ± 0.41	n.s.
Fe (μg/dL)	101.3 ± 25.2	84.0 ± 33.8	0.0001

Abbreviations: BMI, body mass index; SBP, systolic blood pressure; RBC, red blood cell; Plt, platelet; TP, total protein; Alb, albumin; HbA1c, hemoglobin A1c; T-Cho, total cholesterol; TG, triglyceride; AST, aspartate aminotransaminase; ALT, alanine aminotransferase; GGT, gamma glutamyl transpeptidase; Cre, creatinine; Fe, iron; n.s., not significant.

**Table 2 nutrients-14-03453-t002:** Histological evaluation data of biopsy-confirmed NAFLD patients.

Histology (Score)	Number
Steatosis (0/1/2/3)	2/25/6/4
Inflammation (0/1/2/3)	0/22/14/1
Ballooning hepatocyte (0/1/2)	8/19/10
NAS total (2/3/4/5/6)	6/7/13/10/1
Fibrosis stage (0/1/2/3/4)	1/8/14/9/5

**Table 3 nutrients-14-03453-t003:** Comparison of nutrient intake data (NAFLD group, control group).

Parameter	NAFLD	Control	*p* Value	Recommended Nutrient Intake
Total energy intake (kcal)	1864.6 ± 504.4	1915 ± 581	n.s.	1916 ± 302
Carbohydrate (g/day)	237.0 ± 68.9	248.7 ± 80.1	n.s.	302 ± 50
Fat (g/day)	67.5 ± 20.7	61.2 ± 27.6	n.s.	53 ± 8
Protein (g/day)	71.5 ± 19.8	72.2 ± 25.0	n.s.	57 ± 7
Sodium (mg/day)	3641.3 ± 1498.0	3958 ± 1549	n.s.	
Potassium (mg/day)	2127.8 ± 631.7	2350 ± 951	0.0406	2784 ± 199
Calcium (mg/day)	454.1 ± 164.6	498 ± 267	n.s.	695 ± 52
Magnesium (mg/day)	234.3 ± 71.1	255 ± 99	n.s.	324 ± 41
Phosphorus (mg/day)	949.1 ± 242.5	1012 ± 361	n.s.	892 ± 100
Iron (mg/day)	8.03 ± 3.36	7.9 ± 3.3	n.s.	7.3 ± 0.9
Zinc (mg/day)	7.45 ± 2.04	8.4 ± 3.3	0.0091	9 ± 1.5
Copper (mg/day)	1.01 ± 0.27	1.14 ± 0.43	0.0057	0.8 ± 0.1
SFA (g/day)	16.6 ± 6.8	17.9 ± 9.4	n.s.	14.9 ± 2.4
mUSFA (g/day)	22.7 ± 8.5	22.5 ± 11.4	n.s.	
pUSFA (g/day)	13.4 ± 8.3	13.1	Incalculable *	
Cholesterol (mg/day)	342.3 ± 116.3	340 ± 197	n.s.	
Vitamin A (μgRAE/day)	433 ± 174	547 ± 1106	0.0009	789 ± 100
Vitamin B1 (mg/day)	1.04 ± 0.41	0.95 ± 0.47	n.s.	1.2 ± 0.1
Vitamin B2 (mg/day)	1.15 ± 0.52	1.19 ± 0.53	n.s.	1.4 ± 0.2
Vitamin B6 (mg/day)	1.24 ± 0.46	1.20 ± 0.53	n.s.	1.3 ± 0.2
Vitamin B12 (μg/day)	6.31 ± 3.48	6.5 ± 7.9	n.s.	2.4 ± 0
Vitamin C (mg/day)	86.1 ± 47.5	99 ± 73	n.s.	100 ± 0
Vitamin D (μg/day)	6.63 ± 3.96	7.2 ± 8.9	n.s.	8.5 ± 0
Vitamin E (mg/day)	8.02 ± 2.76	6.9 ± 3.5	0.0187	6 ± 0.5
Vitamin K2 (μg/day)	193.9 ± 116.6	250 ± 188	0.0064	150 ± 0
Niacin (mgNE/day)	25.8 ± 8.2	31.3 ± 12.3	0.0002	11 ± 4.7
Folate acid (μg/day)	266.4 ± 105.5	302 ± 169	0.049	240 ± 0
Pantothenic acid (mg/day)	5.87 ± 3.33	5.65 ± 2.19	n.s.	5 ± 0.4
Soluble dietary fiber (g/day)	4.30 ± 1.83	3.6 ± 1.8	0.032	
Insoluble dietary fiber (g/day)	9.61 ± 3.23	11.8 ± 5.2	0.0002	
Total dietary fiber (g/day)	16.8 ± 5.79	18.8 ± 7.3	0.0474	19 ± 1.6
Salt (g/day)	9.4 ± 3.7	10.1 ± 3.9	n.s.	7 ± 0.5

* *p* value was not calculable due to impaired data of control subjects.

**Table 4 nutrients-14-03453-t004:** Comparison of clinical data (early fibrosis NAFLD group, advanced fibrosis NAFLD group).

Parameter	Early Fibrosis (*n* = 23)	Advanced Fibrosis (*n* = 14)	*p* Value
Age (y.o.)	54.6 ± 11.7	56.2 ± 9.6	n.s.
Gender (f/m)	12/11	8/6	n.s.
BMI	34.5 ± 5.9	30.2 ± 5.7	n.s.
SBP (mmHg)	128 ± 13.8	131.6 ± 12.2	n.s.
RBC (×10^4^/mm^3^)	489.1 ± 59.8	464.9 ± 40.6	n.s.
Plt (×10^4^/mm^3^)	232.7 ± 78.2	168.6 ± 81.1	0.0256
TP (g/dL)	7.38 ± 0.43	7.28 ± 0.51	n.s.
Alb (g/dL)	4.3 ± 0.3	4.2 ± 0.4	n.s.
HbA1c (%)	6.90 ± 0.96	6.32 ± 0.73	0.0479
T-Cho (mg/dL)	195.1 ± 40.8	181 ± 23.4	n.s.
TG (mg/dL)	217.3 ± 153.8	139.3 ± 50.1	0.0325
AST (U/L)	51.3 ± 27.8	51.6 ± 8.7	n.s.
ALT (U/L)	66.3 ± 36.8	54.7 ± 41.8	n.s.
GGT (U/L)	131.9 ± 118.8	80.4 ± 44.3	n.s.
Cre (mg/dL)	0.79 ± 0.23	0.74 ± 0.25	n.s.
Fe (μg/dL)	103.3 ± 36.0	129.8 ± 43.4	n.s.

**Table 5 nutrients-14-03453-t005:** Comparison of nutrient intake data (early fibrosis NAFLD group, advanced fibrosis NAFLD group).

Parameter	Early Fibrosis	Advanced Fibrosis	*p* Value
Total energy intake (kcal)	1970 ± 547.0	1692 ± 383.0	n.s.
Carbohydrate (g/day)	245.3 ± 76.5	223.3 ± 53.9	n.s.
Fat (g/day)	73.0 ± 21.4	58.5 ± 16.6	0.0281
Protein (g/day)	76.4 ± 20.3	63.4 ± 16.6	0.0412
Sodium (mg/day)	3959 ± 1687	3119 ± 963.0	n.s.
Potassium (mg/day)	2268 ± 651.7	1898 ± 542.5	n.s.
Calcium (mg/day)	514.4 ± 153.9	355.1 ± 134.2	0.0024
Magnesium (mg/day)	264.2 ± 59.8	214.8 ± 85.3	n.s.
Phosphorus (mg/day)	1016 ± 245.0	838.6 ± 200.1	0.0222
Iron (mg/day)	8.89 ± 3.80	6.62 ± 1.87	0.0208
Zinc (mg/day)	7.87 ± 2.21	6.76 ± 1.55	n.s.
Copper (mg/day)	1.07 ± 0.29	0.90 ± 0.20	0.0475
SFA (g/day)	18.4 ± 7.4	13.6 ± 4.3	0.0174
mUSFA (g/day)	24.8 ± 9.5	19.3 ± 5.0	0.0273
pUSFA (g/day)	13.6 ± 4.8	12.9 ± 12.3	n.s.
Cholesterol (mg/day)	365.1 ± 131.4	304.8 ± 76.2	n.s.
Vitamin A (μgRAE/day)	480.0 ± 189.3	355.7 ± 112.1	0.0169
Vitamin B1 (mg/day)	1.09 ± 0.43	0.96 ± 0.36	n.s.
Vitamin B2 (mg/day)	1.18 ± 0.33	1.10 ± 0.74	n.s.
Vitamin B6 (mg/day)	1.39 ± 0.48	0.98 ± 0.26	0.0018
Vitamin B12 (μg/day)	6.62 ± 3.80	5.80 ± 3.00	n.s.
Vitamin C (mg/day)	88.9 ± 48.4	81.4 ± 47.4	n.s.
Vitamin D (μg/day)	7.30 ± 4.00	5.53 ± 3.79	n.s.
Vitamin E (mg/day)	8.94 ± 3.00	6.52 ± 1.40	0.0022
Vitamin K2 (μg/day)	200.3 ± 114.3	183.4 ± 123.9	n.s.
Niacin (mgNE/day)	27.5 ± 8.2	22.9 ± 7.5	n.s.
Folate acid (μg/day)	278.3 ± 100.5	246.8 ± 114.2	n.s.
Pantothenic acid (mg/day)	6.80 ± 3.91	4.34 ± 0.92	0.0077
Soluble dietary fiber (g/day)	4.78 ± 2.12	3.52 ± 0.79	0.0152
Insoluble dietary fiber (g/day)	10.20 ± 3.62	8.64 ± 2.24	n.s.
Total dietary fiber (g/day)	17.74 ± 6.54	15.34 ± 4.09	n.s.
Salt (g/day)	10.32 ± 4.16	8.01 ± 2.38	0.0394

**Table 6 nutrients-14-03453-t006:** Comparison of clinical data (BMI Low group, BMI High group).

Parameter	BMI Low (*n* = 19)	BMI High (*n* = 18)	*p* Value
Age (y.o.)	61.0 ± 7.6	49.1 ± 10.5	<0.005
BMI	26.9 ± 2.3	35.3 ± 5.2	<0.0001
SBP (mmHg)	130 ± 14.5	129 ± 12.1	n.s.
RBC (×10^4^/mm^3^)	468 ± 60.4	492 ± 44.9	n.s.
Plt (×10^4^/mm^3^)	213 ± 98.0	204 ± 69.5	n.s.
TP (g/dL)	7.32 ± 0.36	7.37 ± 0.55	n.s.
Alb (g/dL)	4.32 ± 0.27	4.25 ± 0.40	n.s.
HbA1c (%)	6.75 ± 0.96	6.60 ± 0.88	n.s.
T-Cho (mg/dL)	187 ± 26.9	193 ± 43.5	n.s.
TG (mg/dL)	199 ± 168	176 ± 74.0	n.s.
AST (U/L)	53.1 ± 35.8	49.6 ± 20.8	n.s.
ALT (U/L)	59.4 ± 44.7	64.5 ± 32.1	n.s.
GGT (U/L)	115 ± 112	110 ± 88.3	n.s.
Cre (mg/dL)	0.72 ± 0.24	0.83 ± 0.23	n.s.

**Table 7 nutrients-14-03453-t007:** Comparison of nutrient intake data (BMI Low group, BMI High group).

Parameter	BMI Low	BMI High	*p* Value
Total calorie (kcal)	1710 ± 506.5	2026 ± 461.2	0.0548
Carbohydrate (g/day)	221 ± 69.1	254 ± 66.4	n.s.
Fat (g/day)	60.3 ± 21.2	75.2 ± 17.7	0.0261
Protein (g/day)	66.2 ± 18.3	77.0 ± 20.3	n.s.
SFA (g/day)	14.3 ± 7.20	19.0 ± 5.5	0.0327
mono USFA (g/day)	20.8 ± 8.9	24.7 ± 7.8	n.s.
poly USFA (g/day)	11.7 ± 4.49	15.2 ± 10.8	n.s.
Cholesterol (mg/day)	306 ± 105	381 ± 118	0.0504
Na (mg/day)	3427 ± 1422	3867 ± 1583	n.s.
K (mg/day)	2212 ± 666.9	2039 ± 598.1	n.s.
Ca (mg/day)	432 ± 155	478 ± 176	n.s.
Mg (mg/day)	235 ± 62.6	233 ± 80.9	n.s.
Phosphorus (mg/day)	908 ± 247	992 ± 237	n.s.
Fe (mg/day)	8.28 ± 4.25	7.77 ± 2.17	n.s.
Zn (mg/day)	6.93 ± 1.98	8.00 ± 2.01	n.s.
Cu (mg/day)	0.99 ± 0.28	1.03 ± 0.27	n.s.
Vitamin A (μgRAE/day)	444 ± 196	422 ± 152	n.s.
Vitamin B1 (mg/day)	0.99 ± 0.48	1.09 ± 0.37	n.s.
Vitamin B2 (mg/day)	1.02 ± 0.26	1.29 ± 0.67	n.s.
Vitamin B6 (mg/day)	1.21 ± 0.48	1.26 ± 0.44	n.s.
Vitamin B12 (μg/day)	6.35 ± 3.00	6.27 ± 4.02	n.s.
Vitamin C (mg/day)	88.2 ± 43.5	83.8 ± 52.6	n.s.
Vitamin D (μg/day)	6.77 ± 3.84	6.47 ± 4.20	n.s.
Vitamin E (mg/day)	8.28 ± 2.88	7.76 ± 2.68	n.s.
Vitamin K2 (μg/day)	211 ± 129	175 ± 103	n.s.
Niacin (mgNE/day)	23.9 ± 6.6	27.8 ± 9.28	n.s.
Folate acid (μg/day)	267 ± 98.5	285 ± 115	n.s.
Pantothenic acid (mg/day)	5.46 ± 3.27	6.31 ± 3.44	n.s.
Water-soluble dietary fiber (g/day)	4.10 ± 1.70	4.52 ± 1.98	n.s.
Insoluble dietary fiber (g/day)	9.32 ± 3.70	9.91 ± 2.71	n.s.
Total dietary fiber (g/day)	16.0 ± 6.04	17.7 ± 5.56	n.s.
Salt (g/day)	8.84 ± 3.52	10.1 ± 3.93	n.s.

**Table 8 nutrients-14-03453-t008:** Comparison of the ratio of each nutrient to total caloric intake (BMI Low group, BMI High group).

Parameter	BMI Low	BMI High	*p* Value
Carbohydrate (g/day/kcal)	13.0 ± 1.5	12.4 ± 1.4	n.s.
Lipid (%)	3.49 ± 0.45	3.74 ± 0.44	n.s.
Protein (%)	3.91 ± 0.51	3.84 ± 0.77	n.s.
SFA (g/day/kcal)	8.09 ± 1.97	9.41 ± 1.88	0.031
mono USFA (g/day/kcal)	11.9 ± 2.46	12.1 ± 1.88	n.s.
poly USFA (g/day/kcal)	6.86 ± 2.03	7.56 ± 4.98	n.s.
Cholesterol (mg/day/kcal)	0.18 ± 0.045	0.20 ± 0.70	n.s.
Na (mg/day/kcal)	1.96 ± 0.47	1.89 ± 0.55	n.s.
K (mg/day/kcal)	1.32 ± 0.30	1.01 ± 0.24	0.0043
Ca (mg/day/kcal)	0.26 ± 0.087	0.24 ± 0.094	n.s.
Mg (mg/day/kcal)	0.14 ± 0.028	0.12 ± 0.037	0.0062
Phosphorus (mg/day/kcal)	0.54± 0.076	0.50 ± 0.097	n.s.
Fe (mg/day/kcal)	5.02 ± 2.83	3.96 ± 1.37	0.0265
Zn (mg/day/kcal)	4.10 ± 0.63	3.97 ± 0.63	n.s.
Cu (mg/day/kcal)	0.58 ± 0.082	0.51 ± 0.10	0.0068
VitA (μgRAE /day/kcal)	0.26 ± 0.12	0.22 ± 0.087	n.s.
Vit B1 (μg/day/kcal)	0.59 ± 0.24	0.53 ± 0.12	n.s.
Vit B2 (μg/day/kcal)	0.61 ± 0.098	0.63 ± 0.27	n.s.
Vit B6 (ng/day/kcal)	0.73 ± 0.28	0.64 ± 0.24	n.s.
Vit B12 (μg/day/kcal)	3.69 ± 1.34	3.04 ± 1.86	n.s.
Vit C (mg/day/kcal)	52.7 ± 24.1	41.4 ± 23.4	n.s.
VitD (mg/day/kcal)	4.06 ± 2.31	3.14 ± 1.72	n.s.
Vit E (μg/day/kcal)	4.86 ± 1.00	3.83 ± 0.87	0.0062
Vit K2 (μg/day/kcal)	0.122 ± 0.060	0.085 ± 0.043	0.0596
Niacin (mgNE /day/kcal)	14.2 ± 2.56	13.7 ± 3.55	n.s.
Folate acid (μg/day/kcal)	0.16 ± 0.052	0.13 ± 0.052	n.s.
Pantothenic acid (mg/day/kcal)	3.19 ± 1.49	3.29 ± 2.37	n.s.
Water-soluble dietary fiber (mg/day/kcal)	2.42 ± 0.77	2.26 ± 0.80	n.s.
Insoluble dietary fiber (g/day/kcal)	5.44 ± 1.57	4.91 ± 0.97	n.s.
Salt (g/day/kcal)	5.07 ± 1.13	4.95 ± 1.35	n.s.

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
