# Peer review of "Characterization of Nutrient Intake in Biopsy-Confirmed NAFLD Patients"

_nutrients, 2022, doi:10.3390/nu14173453_

Round 1

Reviewer 1 Report

Thank you for this interesting paper. My main comment relates to;

1. The potential confounding of including individuals with prediabetes and diabetes and the changes in diet secondary to the disease, whether in table 1 you are describing the HbA1c, it is not clear how many of these individuals had a diabetes diagnosis, and owing the correlation of NALFD with prediabetes and diabetes it will be informative to stratify by HbA1c levels (if possible).

2. I do not think using means and S.D. is suitable for low sample size, median and interquartile range may be more appropriate. Moreover, it is not clear if the tool to record 7-day diet is validated in this population.

3. Please discuss further if diabetes or obesity-related lifestyle may be confounding for the differences observed between NALFD and healthy controls. The conclusions discussed may be biased due to comparing very diseased individuals (i.e. fibrosis) already on treatment and healthy individuals.

4. Merge tables 5a and 5b, and 6a and 6b, and consider moving to supplement.

Thank you.

Author Response

Response to the comments of reviewers

We thank the editor and reviewers for the positive assessment of our manuscript and for identifying areas that required corrections and/or modification. The red-colored text in the revised manuscript is the corrected/modified text. All line numbers mentioned in each response to each comment refer to the small-size numbers that appear on the right margin of the text of the revised manuscript.

Reviewer 1

Thank you for this interesting paper. My main comment relates to;

  1. The potential confounding of including individuals with prediabetes and diabetes and the changes in diet secondary to the disease, whether in table 1 you are describing the HbA1c, it is not clear how many of these individuals had a diabetes diagnosis, and owing the correlation of NAFLD with prediabetes and diabetes it will be informative to stratify by HbA1c levels (if possible).

Thank you for your interesting comment. According to your comment, we investigated the number of T2DM patients in NAFLD subjects. HbA1c ≥ 6.5 % and/or FBS ≥ 126 mg/dl is used to diagnose T2DM. Among 37 NAFLD patients, 21 patients were diagnosed as T2DM. We compared the nutritional intake between non-DM and DM NAFLD patients (Appendix Table). As we demonstrated as following, there are no differences between the two groups. We added some descriptions about this issue in our revised manuscript (line 285-90).

Appendix Table Comparison of nutrient intake data (non-DM NAFLD vs DM NAFLD)

 Parameter

Non-DM (n=16)

DM (n=21)

P value

Total energy intake (kcal)

1,829.8 ± 464.1

1,891.1 ± 542.8

n.s.

Carbohydrate (g/day)

236.5 ± 67.6

237.4 ± 71.4

n.s.

Fat (g/day)

65.4 ± 18.9

69.1 ± 22.3

n.s.

Protein (g/day)

71.3 ± 17.7

71.6 ± 21.7

n.s.

Sodium (mg/day)

3,384.3 ± 1020.8

3,837.2 ± 1,779.0

n.s.

Potassium (mg/day)

2,075.9 ± 652.9

2,167.3 ± 628.3

n.s.

Calcium (mg/day)

425.9 ± 132.1

475.6 ± 186.0

n.s.

Magnesium (mg/day)

243.2 ± 81.7

227.6 ± 63.0

n.s.

Rin (mg/day)

934.6 ± 202.1

960.2 ± 273.7

n.s.

Iron (mg/day)

7.79 ± 1.87

8.22 ± 4.20

n.s.

Zinc (mg/day)

7.21 ± 2.04

7.63 ± 2.07

n.s.

Copper (mg/day)

0.97 ± 0.27

1.02 ± 0.28

n.s.

SFA (g/day)

14.9 ± 6.00

17.86 ± 7.21

n.s.

mUSFA (g/day)

21.2 ± 8.8

23.8 ± 8.3

n.s.

pUSFA (g/day)

14.8 ± 11.8

12.3 ± 3.8

n.s.

Cholesterol (mg/day)

326.7 ± 84.9

354.1 ± 136.3

n.s.

Vitamin A (μg/day)

420.7 ± 125.7

442.4 ± 205.5

n.s.

Vitamin B1 (mg/day)

1.03 ± 0.38

1.05 ± 0.43

n.s.

Vitamin B2 (mg/day)

1.14 ± 0.67

1.15 ± 0.35

n.s.

Vitamin B6 (mg/day)

1.20 ± 0.40

1.27 ± 0.50

n.s.

Vitamin B12 (μg/day)

5.65 ± 2.53

6.81 ± 4.05

n.s.

Vitamin C (mg/day)

83.5 ± 47.0

88.0 ± 49.0

n.s.

Vitamin D (mg/day)

6.63 ± 3.96

7.2 ± 8.9

n.s.

Vitamin E (mg/day)

7.93 ± 2.72

8.10 ± 2.86

n.s.

Vitamin K2 (μg/day)

221.9 ± 145.8

172.6 ± 86.2

n.s.

Niacin (mg/day)

26.8 ± 9.4

25.0 ± 7.1

n.s.

Folate acid (mg/day)

267.4 ± 110.6

265.7 ± 104.2

n.s.

Pantothenic acid (mg/day)

5.85 ± 3.70

5.89 ± 3.12

n.s.

Soluble dietary fiber (g/day)

4.77 ± 2.11

3.95 ± 1.54

n.s.

Insoluble dietary fiber (g/day)

9.94 ± 3.45

9.35 ± 3.11

n.s.

Total dietary fiber (g/day)

17.57 ± 6.32

16.27 ± 5.45

n.s.

Salt (g/day)

8.89 ± 2.44

9.87 ± 4.49

n.s.

  1. I do not think using means and S.D. is suitable for low sample size, median and interquartile range may be more appropriate. Moreover, it is not clear if the tool to record 7-day diet is validated in this population.

Thank you for your valuable comments. We obtained our control data from the National Health and Nutrition Survey (NHNS) database. Because individual information was not available for the nutrition survey, t-values and degrees of freedom were calculated based on aggregate information of mean, standard deviation, and number of subjects as shown in the formula below [1] [2].

As you pointed our number of study subjects (n=37) was relatively small, we would like to use mean and S.D. for the comparison of our study subject data with NHNS control data. In NHNS control data, individual information was not available, and we could not get 7-day diet record data from control subjects.

  1. Please discuss further if diabetes or obesity-related lifestyle may be confounding for the differences observed between NAFLD and healthy controls. The conclusions discussed may be biased due to comparing very diseased individuals (i.e. fibrosis) already on treatment and healthy individuals.

Thank you for your important comments. We compared the nutritional intake data between non-DM and DM NAFLD subjects as described above (Appendix Table). We added some descriptions about this issue in our revised manuscript (line 285-90). 

  1. Merge tables 5a and 5b, and 6a and 6b, and consider moving to supplement.

Thank you for your constructive comments. We moved Table 5a, 5b, 6a, and 6b to supplement files.

References

  1. Welch, B. L., The significance of the difference between two means when the population variances are unequal. Biometrika 1938, 29, (3/4), 350-362.
  2. Satterthwaite, F. E., An approximate distribution of estimates of variance components. Biometrics bulletin 1946, 2, (6), 110-114.

Reviewer 2 Report

Dear authors this is a very interesting study on the Relationship between nutrient intake and liver disease progression in biopsy-confirmed NAFLD patients. The article is well-written. However, I have several remarks and suggestion:

1. In my opinion the article title does not fully correspond to the content. The relatively lower nutrients intake among patients with NAFLD might be attributed to dietary changes because of the condition and might not be used as an indicator of the condition’s progress.

2. The Abstract needs improvement in several points:

- Methods are not clearly described. The fact that 37 out of 5074 NHNS participants were used as controls is not mentioned and is not clear. The same about the different procedure used for the nutrients intake evaluation among patients and controls. Recorded dietary intake (please remove of nutrients)

- Results: excesses and deficiencies in comparison with what?

- Conclusions: In my opinion the conclusion is that differences were found in nutrients intake of patients and controls and according to the severity of the condition, BMI etc. Your results do not support that biases (preferably discrepancies, differences) in nutrients intake could contribute to NAFLD.

Methods: please describe the procedures used for the evaluation of controls dietary intake. Figure 1: please translate in English.

Results: Did you compare your findings with Japanese dietary guidelines for nutrients intake? If yes, were they in the recommended range? Only if they do not fall in this range, only then it can be mentioned that there are excesses and deficiencies.

Discussion:  Limitations 2, 4 and 5 are really serious. In my opinion limitation 1 in your manuscript does not consist a limitation.

Conclusions: In the first sentence please consider removing the last part (…, which may contribute to fibrosis progression). I disagree with the use of the word bias (please see above). In my opinion the last sentence should be removed.

Author Response

Response to the comments of reviewers

We thank the editor and reviewers for the positive assessment of our manuscript and for identifying areas that required corrections and/or modification. The red-colored text in the revised manuscript is the corrected/modified text. All line numbers mentioned in each response to each comment refer to the small-size numbers that appear on the right margin of the text of the revised manuscript.

Reviewer 2

Dear authors this is a very interesting study on the Relationship between nutrient intake and liver disease progression in biopsy-confirmed NAFLD patients. The article is well-written. However, I have several remarks and suggestion:

  1. In my opinion the article title does not fully correspond to the content. The relatively lower nutrients intake among patients with NAFLD might be attributed to dietary changes because of the condition and might not be used as an indicator of the condition’s progress.

Thank you for your nice suggestion. I agree to your suggestion. According to your suggestion, we changed our title to “Character of nutrient intake in biopsy-confirmed NAFLD patients”.

  1. The Abstract needs improvement in several points:

- Methods are not clearly described. The fact that 37 out of 5074 NHNS participants were used as controls is not mentioned and is not clear. The same about the different procedure used for the nutrients intake evaluation among patients and controls. Recorded dietary intake (please remove of nutrients)

I make the mistake of incorrectly describing the method. We have checked with our co-author (M.F.), a statistical expert, and found the following statement to be correct. We corrected the descriptions in our revised manuscript as followings.

In this test, we are directly comparing the data of 5,074 people of the same age and sex as the 37 respondents. This test is a direct comparison, not a matching (line 121-3).

Also, we changed the descriptions in Abstract as you pointed (line 50-1). Thank you very much for your important comments.

- Results: excesses and deficiencies in comparison with what?

Thank you for your comments. According to your comments, we checked our manuscript and add some descriptions in our revised manuscript abstract (line 55, 58).

- Conclusions: In my opinion the conclusion is that differences were found in nutrients intake of patients and controls and according to the severity of the condition, BMI etc. Your results do not support that biases (preferably discrepancies, differences) in nutrients intake could contribute to NAFLD.

Thank you for your valuable comments. According to your comments, we changed the descriptions in abstract conclusions and main text (line 60-1, 311-12).

Methods: please describe the procedures used for the evaluation of controls dietary intake. Figure 1: please translate in English.

Thank you for your comments. We used the food ingredients list frequency survey sheet written in Japanese in this study. We translate the sheet into English in our revised manuscript (Figure 1a).

Results: Did you compare your findings with Japanese dietary guidelines for nutrients intake? If yes, were they in the recommended range? Only if they do not fall in this range, only then it can be mentioned that there are excesses and deficiencies.

Thank you for your comments. We compared our study subject data (control, NAFLD patients) with Japanese guidelines for nutrient intake (https://www.mhlw.go.jp/stf/newpage_22536.html). Even in control subjects, the nutrient intake volume was a little smaller than recommendation nutrient intake in some nutrients. In NAFLD patients, nutrition intake was more deficient or excessive than control subjects. So, there are excesses and deficiencies in NAFLD patients in our study. We added some descriptions about this issue in our revised manuscript (line 239-41).

Discussion:  Limitations 2, 4 and 5 are really serious. In my opinion limitation 1 in your manuscript does not consist a limitation.

Thank you for your kind comments. Limitation 1 was recommended to describe by another reviewer. We also think limitations 2, 4, and 5 are very important. Therefore, we described 5 limitations in our manuscript. Thank you.

Conclusions: In the first sentence please consider removing the last part (…, which may contribute to fibrosis progression). I disagree with the use of the word bias (please see above). In my opinion the last sentence should be removed.

Thank you again for your critical and important comments. According to your comments, we changed the descriptions in abstract conclusions and main text (line 60-1, 311-12).

Round 2

Reviewer 1 Report

Thank you for the corrections submitted, I have two comments regarding the changes that are still missing:

1. The change of the title is not explained, the study was mainly descriptive thus, I would use the verbal form to 'characterization'.

2. In lines 329 to 335 it is unclear what is the meaning of FBS, moreover, according to whom was the definition of T2D adopted (WHO, ADA)? did patients clinically diagnosed with T2D according to ICD? thus, if no proper diagnosis has been established, I would discourage the use of the clinical term 'T2D' and just maintain the category of those above the cutoff values for HbA1c and fasting glucose.

Thank you

Author Response

Response to the comments of reviewers

We thank the editor and reviewers for the positive assessment of our manuscript and for identifying areas that required corrections and/or modification. The red-colored text in the revised manuscript is the corrected/modified text. All line numbers mentioned in each response to each comment refer to the small-size numbers that appear on the right margin of the text of the revised manuscript.

Reviewer 1

Thank you for the corrections submitted, I have two comments regarding the changes that are still missing:

  1. The change of the title is not explained, the study was mainly descriptive thus, I would use the verbal form to 'characterization'.

We apologize for the lack of explanation about the change of the title. Reviewer 2 advised us to change title to correspond to our study content. As you pointed, I also think “characterization” is better verbal form. We changed our title to ”Characterization of nutrient intake in biopsy-confirmed NAFLD patients”. Thank you for your nice suggestion.

  1. In lines 329 to 335 it is unclear what is the meaning of FBS, moreover, according to whom was the definition of T2D adopted (WHO, ADA)? did patients clinically diagnosed with T2D according to ICD? thus, if no proper diagnosis has been established, I would discourage the use of the clinical term 'T2D' and just maintain the category of those above the cutoff values for HbA1c and fasting glucose.

Thank you for your important comment. We diagnosed T2DM according to the Japanese clinical practice guideline for diabetes. We added some descriptions and a reference (no. 35) in our revised manuscript (line 290-2). In addition, we corrected the error in the unit description after consulting with the dietitian in Tables.

Reviewer 2 Report

Dear authors thank you for considering and applying our suggestions to your manuscript. In general I believe that the revised manuscript is significantly improved.

In my opinion one more table should be included with the recommended nutrients intake for this population group and the nutrients intake of cases and controls. Furthermore, the quality of Figure 1a should be improved. Unfortunately, it is not easily readable.

Author Response

Response to the comments of reviewers

We thank the editor and reviewers for the positive assessment of our manuscript and for identifying areas that required corrections and/or modification. The red-colored text in the revised manuscript is the corrected/modified text. All line numbers mentioned in each response to each comment refer to the small-size numbers that appear on the right margin of the text of the revised manuscript.

Reviewer 2

Dear authors thank you for considering and applying our suggestions to your manuscript. In general I believe that the revised manuscript is significantly improved.

In my opinion one more table should be included with the recommended nutrients intake for this population group and the nutrients intake of cases and controls. Furthermore, the quality of Figure 1a should be improved. Unfortunately, it is not easily readable.

Thank you for your kind and valuable comments. According to your comment, we added one column which demonstrated recommended nutrient intake in Japanese guideline in Table 2. We added some descriptions in the revised manuscript (line 191-5).

As you pointed, Figure 1a is not easily readable due to the small size. We added supplementary figure 1 (original size). In addition, we corrected the error in the unit description after consulting with the dietitian in Tables.

This manuscript is a resubmission of an earlier submission. The following is a list of the peer review reports and author responses from that submission.

Round 1

Reviewer 1 Report

The authors described detailed nutrient intake of biopsy-proven NAFLD patients with an elaborate review of each diet. The study results demonstrated that specific nutrients are excessive or deficient in NAFLD patients, implying the valuable dietary suggestion for these patients. 

A few points need to be addressed. 

1. I strongly recommend review/revision by a native English speaker to make everything as clear as possible.

2. Please concisely address the investigation method of the National Institute of Health and Nutrition. I recommend describing why the authors set up the control group from national data, instead of selecting healthy volunteers as a control group, and why the authors studied a relatively small number of patients, rather than prolonging the study period and enrolling more patients. 

3. Please address the indication of liver biopsy in Materials and Methods, whether it is routinely performed for the diagnosis or only in certain selected cases. 

4. It would be interesting to address a dietary questionnaire and photographs of each diet as an example of supplementary data. 

5. Please address the discussion for the results of nutrient intake of the early and advanced fibrosis NAFLD group. 

6. Please address the number of patients and control groups in each table. 

7. The annotation for the p-value in each table does not seem to be necessary. 

Author Response

Response to the comments of reviewers

We thank the editor and reviewers for the positive assessment of our manuscript and for identifying areas that required corrections and/or modification. The red-colored text in the revised manuscript is the corrected/modified text. All line numbers mentioned in each response to each comment refer to the small-size numbers that appear on the right margin of the text of the revised manuscript.

Reviewer 1

The authors described detailed nutrient intake of biopsy-proven NAFLD patients with an elaborate review of each diet. The study results demonstrated that specific nutrients are excessive or deficient in NAFLD patients, implying the valuable dietary suggestion for these patients. 

A few points need to be addressed. 

  1. I strongly recommend review/revision by a native English speaker to make everything as clear as possible.

Thank you for your comments. According to your comments, we received English editing (line 307-8) and corrected our manuscript carefully.

  1. Please concisely address the investigation method of the National Institute of Health and Nutrition. I recommend describing why the authors set up the control group from national data, instead of selecting healthy volunteers as a control group, and why the authors studied a relatively small number of patients, rather than prolonging the study period and enrolling more patients. 

Thank you for your important comments. We selected fully age- and gender-matched cases from the National Institute of Health and Nutrition in 2019 to serve as controls. According to your comments, we added some descriptions in our revised manuscript (line 124-5). The National Health and Nutrition Survey (NHNS) was used as a control group for comparison. The purpose of this survey was to clarify citizens’ physical conditions, nutrient intake, and lifestyle based on the Health Promotion Law (Law No.103, enacted in 2002), and to get basic data to comprehensively promote people’s health. It comprises three parts: physical examination, dietary survey, and lifestyle habits questionnaire, and the surveys are conducted by medical personnel including dietitians, physicians, nurses, public health nurses, and clinical laboratory technicians. This survey in 2019 is a survey of all households (1 year old and over) in 296 areas, excluding 4 areas that could not be surveyed because of the effects of typhoons, among the areas randomly selected from the general areas of the census. This survey, which uses cluster sampling to select and conduct a full survey, has more external validity than using healthy volunteers as the control group because it includes a wider range of subjects, including latent conditions. 

Biopsy-confirmed NAFLD patients were enrolled in 2020 during the registration period in this study from three hepatology centers in Japan. In order to elucidate the recent nutritional intake of NAFLD patients, we set study registration period half year. So the sample size was relatively small as you pointed. In this study we found some significant results from this cohort, we think this sample size was enough to investigate nutritional intake of NAFLD patients in our study. We added some descriptions about this issue to our revised manuscript (line 102-7).

  1. Please address the indication of liver biopsy in Materials and Methods, whether it is routinely performed for the diagnosis or only in certain selected cases. 

Thank you for your valuable comments. The indication of liver biopsy was routinely performed for the diagnosis of liver histology in each center. According to your comments, we added some descriptions in our revised manuscript (line 105-7).

  1. It would be interesting to address a dietary questionnaire and photographs of each diet as an example of supplementary data. 

Thank you for your nice suggestion. Our dietary questionnaire was written in Japanese as below. We added the sample of dietary questionnaire sheet and photographs in Figure 1. We added some descriptions and Figure 1in our revised manuscript (line 134-50, Figure 1).

Figure 1

  1. a) The food ingredients list frequency survey sheet

  1. b) Sample photographs of NAFLD patients in one day

  1. Please address the discussion for the results of nutrient intake of the early and advanced fibrosis NAFLD group. 

Thank you for your critical comments. According to your comments, we added some descriptions in Discussion (line 260-1, 267-71).

  1. Please address the number of patients and control groups in each table. 

Thank you for your comments. According to your comments, we added the number of the study subjects in each table.

  1. The annotation for the p-value in each table does not seem to be necessary. 

Thank you for your nice comments. According to your comments, we deleted the annotations for the p value in each table.

Reviewer 2 Report

Kamada et al. assessed the nutritional intake of biopsy-proven NAFLD patients and compared this data to data from healthy participants obtained from the National Institute of Health and Nurition. Moreover, they separated their data sets and analysed it by sex, BMI, or fibrosis stage. Unfortunately, especially the methodological part of the manuscript is not sufficiently described. It is not clear how the dietary data were evaluated (on which the entire manuscript is based on) and how comparable the control cohort is. Please see below for details.

The methods evaluating the nutritional intake of these patients are poorly described: Which dietary questionnaire was used? FFQ? Why did authors decide using a questionnaire instead of a food diary? Which food groups were queried? Which software was used for assessing nutritional intake? Which food-data base was used? How were the pictures analysed? How were that data from questionnaire and pictures linked?

Moreover, it is not clear how the nutrition of the control group was assessed and analysed? If data sets are compared they need to use the same method for assessing nutritional intake as well as the same software for analysing this data.

The age of the study subjects is not mentioned.

There are several recently published studies demonstrating that dietary habits changed during covid-19 pandemic and several lockdowns. As the control data was obtained in 2019, before the pandemic, it seems that data from NAFLD patients were obtained during the pandemic? This could have an effect on the outcomes, the authors need to comment on this.

Underreporting is a common issue in dietary investigationss and need to be determined, especially in a study cohort with metabolic abnormalities (see e.g. Rosell et al. Am J Clin NutI' 2003).

Author Response

Response to the comments of reviewers

We thank the editor and reviewers for the positive assessment of our manuscript and for identifying areas that required corrections and/or modification. The red-colored text in the revised manuscript is the corrected/modified text. All line numbers mentioned in each response to each comment refer to the small-size numbers that appear on the right margin of the text of the revised manuscript.

Reviewer 2

Kamada et al. assessed the nutritional intake of biopsy-proven NAFLD patients and compared this data to data from healthy participants obtained from the National Institute of Health and Nurition. Moreover, they separated their data sets and analysed it by sex, BMI, or fibrosis stage. Unfortunately, especially the methodological part of the manuscript is not sufficiently described. It is not clear how the dietary data were evaluated (on which the entire manuscript is based on) and how comparable the control cohort is. Please see below for details.

The methods evaluating the nutritional intake of these patients are poorly described: Which dietary questionnaire was used? FFQ? Why did authors decide using a questionnaire instead of a food diary? Which food groups were queried?

Thank you for your valuable comments. We added Figure 1 and some descriptions in our revised manuscript to describe the methods evaluating the nutritional intake of the study subjects in this study (line 134-50, Figure 1). For the food ingredients list frequency survey sheet used (Figure 1a), we created and used our own sheet for each meal that allows the names of commonly eaten ingredients to be checked based on the classification of the food composition table. This is because it is not always possible to identify an ingredient only by a photograph (Figure 1b) and the name of the menu.

Figure 1

  1. a) The food ingredients list frequency survey sheet

  1. b) Sample photographs of NAFLD patients in one day

Which software was used for assessing nutritional intake? Which food-data base was used?

For the evaluation of nutrient intake, the Food Composition Database of the Ministry of Education, Culture, Sports, Science and Technology (MEXT) was used in this study. We added some descriptions in the manuscript (line145-6).

How were the pictures analysed? How were that data from questionnaire and pictures linked?

Data from photographs taken by patients of everything they ate and drank, with a ruler placed as a guide to determine size (Figure 1b), were checked with the food ingredients list frequency survey sheet, which was checked at each meal, and nutritional calculations were made using the Ministry of Education's food database. In order to ensure a reliable understanding of what was eaten, the photographs of each food and the food ingredients list frequency survey sheet were used together in this study.

Moreover, it is not clear how the nutrition of the control group was assessed and analysed? If data sets are compared they need to use the same method for assessing nutritional intake as well as the same software for analysing this data.

Thank you for your important comments. We selected fully age- and gender-matched cases from the National Institute of Health and Nutrition in 2019 to serve as controls. According to your comments, we added some descriptions in our revised manuscript (line 124-5).

The National Health and Nutrition Survey (NHNS) was used as a control group for comparison. The purpose of this survey was to clarify citizens’ physical conditions, nutrient intake, and lifestyle based on the Health Promotion Law (Law No.103, enacted in 2002), and to get basic data to comprehensively promote people’s health. It comprises three parts: physical examination, dietary survey, and lifestyle habits questionnaire, and the surveys are conducted by medical personnel including dietitians, physicians, nurses, public health nurses, and clinical laboratory technicians. This survey in 2019 is a survey of all households (1 year old and over) in 296 areas, excluding 4 areas that could not be surveyed because of the effects of typhoons, among the areas randomly selected from the general areas of the census. This survey, which uses cluster sampling to select and conduct a full survey, has more external validity than using healthy volunteers as the control group because it includes a wider range of subjects, including latent conditions. 

The age of the study subjects is not mentioned.

Thank you for your comments. According to your comments, we added age data of the study subjects in each table.

There are several recently published studies demonstrating that dietary habits changed during covid-19 pandemic and several lockdowns. As the control data was obtained in 2019, before the pandemic, it seems that data from NAFLD patients were obtained during the pandemic? This could have an effect on the outcomes, the authors need to comment on this.

Thank you for your important comments. We recently investigated the lifestyle changes between before and during COVID19 pandemic (ref. 38). In the study, we found new MAFLD diagnoses and daily alcohol intake increased during pandemic. Although NAFLD patients in the present study did not increase alcohol intake, there might have some additional effects of pandemic on our study results. We added some descriptions in our revised manuscript about COVID-19 pandemic related issue (line 299-304).

Underreporting is a common issue in dietary investigations and need to be determined, especially in a study cohort with metabolic abnormalities (see e.g. Rosell et al. Am J Clin NutI' 2003)

Thank you for your critical comments. Indeed, underreporting is a common and important issue in dietary investigations. We added some descriptions and a reference to our revised manuscript about this issue (line 258, ref. 23).

Reviewer 3 Report

The main limitation of this study is that cases and controls are not well matched. Moreover, it is possible that some people in control groups had NAFLD.

Author Response

Response to the comments of reviewers

We thank the editor and reviewers for the positive assessment of our manuscript and for identifying areas that required corrections and/or modification. The red-colored text in the revised manuscript is the corrected/modified text. All line numbers mentioned in each response to each comment refer to the small-size numbers that appear on the right margin of the text of the revised manuscript.

Reviewer 3

The main limitation of this study is that cases and controls are not well matched. Moreover, it is possible that some people in control groups had NAFLD.

Thank you for your important comments. We selected fully age- and gender-matched cases from the National Institute of Health and Nutrition in 2019 to serve as controls. We added some descriptions to our revised manuscript (line 124-5).

The National Health and Nutrition Survey (NHNS) was used as a control group for comparison. The purpose of this survey was to clarify citizens’ physical conditions, nutrient intake, and lifestyle based on the Health Promotion Law (Law No.103, enacted in 2002), and to get basic data to comprehensively promote people’s health. It comprises three parts: physical examination, dietary survey, and lifestyle habits questionnaire, and the surveys are conducted by medical personnel including dietitians, physicians, nurses, public health nurses, and clinical laboratory technicians. This survey in 2019 is a survey of all households (1 year old and over) in 296 areas, excluding 4 areas that could not be surveyed because of the effects of typhoons, among the areas randomly selected from the general areas of the census. This survey, which uses cluster sampling to select and conduct a full survey, has more external validity than using healthy volunteers as the control group because it includes a wider range of subjects, including latent conditions. 

As you pointed, our control subjects might include NAFLD patients. However, liver-related parameters (AST, ALT, GGT, platelet count) of our control subjects were within normal limit. So we think our control subjects did not include NAFLD patients with severe liver injury at least.

Round 2

Reviewer 2 Report

Kamada et al. revised the manuscript, however, the question below, which determines whether the control data can be used as control data, was still not answered.

“Question 1st Revision: Moreover, it is not clear how the nutrition of the control group was assessed and analysed? If data sets are compared they need to use the same method for assessing nutritional intake as well as the same software for analysing this data.

Answer Authors: Thank you for your important comments. We selected fully age- and gender-matched cases from the National Institute of Health and Nutrition in 2019 to serve as controls. According to your comments, we added some descriptions in our revised manuscript (line 124-5).

The National Health and Nutrition Survey (NHNS) was used as a control group for comparison. The purpose of this survey was to clarify citizens’ physical conditions, nutrient intake, and lifestyle based on the Health Promotion Law (Law No.103, enacted in 2002), and to get basic data to comprehensively promote people’s health. It comprises three parts: physical examination, dietary survey, and lifestyle habits questionnaire, and the surveys are conducted by medical personnel including dietitians, physicians, nurses, public health nurses, and clinical laboratory technicians. This survey in 2019 is a survey of all households (1 year old and over) in 296 areas, excluding 4 areas that could not be surveyed because of the effects of typhoons, among the areas randomly selected from the general areas of the census. This survey, which uses cluster sampling to select and conduct a full survey, has more external validity than using healthy volunteers as the control group because it includes a wider range of subjects, including latent conditions.”

Unfortunately, these comments do not explain how diet was collected in the control cohort, only that diet was collected. The nutrition should have been collected with the same questionnaire and analysed with the same software, moreover, the control subjects must also have taken pictures, otherwise comparability is difficult.

Figure 1a is not readable and in a foreign language.

As underreporting is a common and important issue this not only needs to be commented but also analysed based on the data.

Author Response

Response to the comments of reviewers

We thank the editor and reviewers for the positive assessment of our manuscript and for identifying areas that required corrections and/or modification. The red-colored text in the revised manuscript is the corrected/modified text. All line numbers mentioned in each response to each comment refer to the small-size numbers that appear on the right margin of the text of the revised manuscript.

Reviewer 2

Kamada et al. revised the manuscript, however, the question below, which determines whether the control data can be used as control data, was still not answered.

“Question 1st Revision: Moreover, it is not clear how the nutrition of the control group was assessed and analysed? If data sets are compared they need to use the same method for assessing nutritional intake as well as the same software for analysing this data.

Answer Authors: Thank you for your important comments. We selected fully age- and gender-matched cases from the National Institute of Health and Nutrition in 2019 to serve as controls. According to your comments, we added some descriptions in our revised manuscript (line 124-5).

The National Health and Nutrition Survey (NHNS) was used as a control group for comparison. The purpose of this survey was to clarify citizens’ physical conditions, nutrient intake, and lifestyle based on the Health Promotion Law (Law No.103, enacted in 2002), and to get basic data to comprehensively promote people’s health. It comprises three parts: physical examination, dietary survey, and lifestyle habits questionnaire, and the surveys are conducted by medical personnel including dietitians, physicians, nurses, public health nurses, and clinical laboratory technicians. This survey in 2019 is a survey of all households (1 year old and over) in 296 areas, excluding 4 areas that could not be surveyed because of the effects of typhoons, among the areas randomly selected from the general areas of the census. This survey, which uses cluster sampling to select and conduct a full survey, has more external validity than using healthy volunteers as the control group because it includes a wider range of subjects, including latent conditions.”

Unfortunately, these comments do not explain how diet was collected in the control cohort, only that diet was collected. The nutrition should have been collected with the same questionnaire and analysed with the same software, moreover, the control subjects must also have taken pictures, otherwise comparability is difficult.

Thank you for your comments. As your comments, our control data were not collected with the same questionnaire or our control subjects have not taken pictures. We added some descriptions about this issue as limitations in discussion section (line 304-11). Our control data were not collected with the same questionnaire nor our control subjects have not taken pictures. Since the survey method is to sequentially weigh and record dietary contents rather than having the researcher interview and recall them, we think the survey method is equivalent to taking a photograph of each meal. The National Health and Nutrition Survey (NHNS) used in this study as a control group for comparison includes not only nutritional intake but also lifestyle habits and health examinations (blood data). This survey, which uses cluster sampling to select and conduct a full survey, has more external validity than using healthy volunteers as the control group because it includes a wider range of subjects, including latent conditions.

Figure 1a is not readable and in a foreign language.

Thank you for your comments. Since some reviewers recommended to present a dietary questionnaire as an example of data in our manuscript, we inserted the food ingredients list frequency survey sheet sample to Figure 1a. As you pointed this sheet was written in Japanese, so we added annotation in text (line 133).

As underreporting is a common and important issue this not only needs to be commented but also analysed based on the data.

Thank you for your valuable comments. According to your comments, we added some descriptions in our manuscript (line 256-9).

Reviewer 3 Report

My comments are addressed.

Author Response

Response to the comments of reviewers

We thank the editor and reviewers for the positive assessment of our manuscript and for identifying areas that required corrections and/or modification. The red-colored text in the revised manuscript is the corrected/modified text. All line numbers mentioned in each response to each comment refer to the small-size numbers that appear on the right margin of the text of the revised manuscript.

Reviewer 3

My comments are addressed.

Thank you for your various comments in revision process.
